# Relative positions of the polar boundary of the outer electron radiation belt and the equatorial boundary of the auroral oval

Maria O. Riazanteseva[1], Elizaveta E. Antonova[1,2], Marina V. Stepanova[3], Boris V.Marjin [2],
Ilia A. Rubinshtein[2], Vera O. Barinova[2], and Nikita V. Sotnikov[2]

[1]Space Research Institute (IKI) Russian Academy of Science, Moscow, Russia
[2]Skobeltsyn Institute of Nuclear Physics, Lomonosov Moscow State University, Moscow, Russia
[3]Physics Department, Universidad de Santiago de Chile (USACH)

**Correspondence:** Maria O. Riazantseva (orearm@gmail.com)

**Abstract.** Finding the position of the polar boundary of the outer electron radiation belt, relative to the position of the auroral oval, is a long-standing problem. Here we analyze it using the data of the METEOR-M1 auroral satellite for the period from 11 November 2009 to 27 March 2010. The geomagnetic conditions during the analyzed period were comparatively quiet. METEOR-M1 has a polar solar-synchronous circular orbit with an altitude of $\approx 832$ km, a period of 101.3 min, and an

inclination of $98°$. We analyze flux observations of auroral electrons with energies between 0.03 and 16 keV, and electrons with energies $> 100$ keV, measured simultaneously by the GGAK-M set of instruments, composed by semiconductors, scintillator detectors, and electrostatic analyzers. We assume that in the absence of geomagnetic storms the polar boundary of the outer radiation belt can be identified as a decrease in the count rate of precipitating energetic electrons to the background level. It was found that this boundary can be located both inside the auroral oval or equatorward of the equatorial boundary of the auroral

precipitation. It was also found that for slightly disturbed geomagnetic conditions the polar boundary of the outer radiation belt is almost always located inside the auroral oval. We observe that the difference between the position of the polar boundary of the outer radiation belt and the position of the equatorial boundary of the auroral precipitation depends on the AE and PC indices of geomagnetic activity. The implications of these results in the analysis of the formation of the outer radiation belt are discussed.

## 1 Introduction

The position of the trapping boundary for energetic electrons in the outer radiation belt (ORB) contains information about the topology of the magnetic field lines of the Earth. For a long time this has been analyzed using data from both low-orbiting and high-apogee satellites (Frank et al., 1964; Frank, 1971; Fritz, 1968, 1970; McDiarmid and Burrows, 1968; Vernov et al., 1969; Imhof et al., 1990, 1991, 1992, 1993; Kanekal et al., 1998, etc.). Using the data of high-apogee satellites, Vernov et al.

(1969) showed that the polar boundary of the ORB, also known as the trapping boundary, is located near to $\sim 9\ R_E$ in the dayside sector and near to $\sim 7-8\ R_E$ close to midnight. These results were further supported by Imhof et al. (1993) using data from the CRRES and SCATHA satellites, and covering distances from $\sim 6$ to $\sim 8.3\ R_E$ (CRRES) and from $\sim 7$ to $\sim 8.5\ R_E$

(SCATHA). Results obtained by (Fritz, 1968, 1970; Imhof et al., 1997; Yahnin et al., 1997) show that the isotropic boundary of energetic particles (i.e. the boundary where pitch-angle of particles becomes isotropic) is located equatorward of the trapping boundary. It means that the ORB trapping boundary can be clearly identifiable using low orbiting satellites measurements.

A good understanding of the relative positions of the trapping boundary and the equatorial edge of the auroral oval is important for the analysis of the structure of magnetospheric plasma domains and the topology of the geomagnetic field. Comparison of the relative position of the trapping boundary and the auroral oval was statistically done using ground-based auroral observations and satellite observations of the trapping boundary. Akasofu (1968) compared the position of Feldstein's auroral oval with the trapping boundary of the 40 keV electrons obtained by Frank et al. (1964) and statistically showed that the trapping boundary is located inside the auroral oval. However, later Feldstein and Starkov (1970) compared the position of the auroral oval with the results of Alouette-2 observations and concluded that the auroral oval is situated just on the polar border of the trapped radiation region of electrons with energy > 35 keV. Rezhenov et al. (1975) analyzed particle fluxes with energies 0.27, 11, 28 and 63 keV, from the COSMOS-424 satellite, and showed that the trapping boundary is located poleward of the region of low energy electron precipitation. However, this study was done using the data obtained for only 21 orbits, and was not widely known. Feldstein et al. (2014) stressed (p. 120 in their paper), that poleward (high-latitude) boundary of the diffuse auroral belt without any discrete auroral forms "constitutes the equatorward boundary of the auroral oval and at the same time it is the high-latitude boundary of the radiation belt (RB) of electrons with energies from a few tens to hundreds of kiloelectronvolts (STB – stable trapping boundary for radiation belt electrons)".

According to the traditional point of view (see, for example, Paschmann et al., 2002), the auroral oval is mapped to the plasma sheet. In this case the trapping boundary should be located equatorward or at the equatorial boundary of the auroral oval. However, Antonova et al. (2014b, 2015), and Kirpichev et al. (2016) showed that most part of the auroral oval does not map to the plasma sheet. It is mapped to the plasma ring surrounding the Earth at geocentric distances from $\sim 7\,R_E$ to the magnetopause, near noon, and to $\sim 10-13\,R_E$ near midnight. They suggested that the plasma in the magnetosphere is in magnetostatic equilibrium, and used the value of plasma pressure as a natural tracer of magnetic field lines, comparing the pressure at low latitudes and at the equatorial plane. Antonova et al. (2017) showed that the outer boundary of this ring in the night sector coincides with the external boundary of the ring current. Results obtained by (Antonova et al., 2014b, 2015, 2017; Kirpichev et al., 2016) showed that the auroral oval is mapped to the region of quasitrapping, where drift trajectories of energetic electrons with pitch-angles smaller than near to 90° surround the Earth (Delcourt and Sauvaud, 1999; Öztürk and Wolf, 2007; Ukhorskiy et al., 2011; Antonova et al., 2011a) due to drift shell splitting effect (which is ordinarily named as Shabansky effect). Such mapping suggests that the trapping boundary should be located poleward of the equatorial boundary of the auroral oval.

Therefore, it is very important to establish the true location of the trapping boundary with respect to the equatorial auroral oval boundary. This can be done using simultaneous observations of both auroral electron precipitation and fluxes of energetic electrons. It is well known that the location of the auroral oval and the location of the trapping boundary are strongly affected by geomagnetic activity. Therefore, it is necessary to compare these relative locations using simultaneous measurements of the auroral oval and trapping boundary on the same satellite. However, there are some difficulties related to the detection of the

trapping boundaries during the periods of low geomagnetic activity (for example during the solar minimum). In these cases, the level of electron fluxes inside the ORB can be rather low, close to the limit of sensitivity of the instrument. Thus, the detected trapping boundary can be located closer equatorward with respect to the true trapping boundary.

Despite the significant amount of particle measurements carried out by low-orbiting satellites, the relative location of the trapping boundary and the equatorial boundary of the auroral oval, and how they could be affected by geomagnetic activity, still requires careful studies. In this work, we use data of the satellite METEOR-M1 to establish the location of the trapping boundary and of the auroral oval for different levels of geomagnetic activity, which were quantified using the AE and PC geomagnetic indices. The paper is organized as follows. First, we describe the METEOR-M1 satellite instrumentation and the data analysis, including important caveats. Then we obtain the position of the trapping boundary of electrons with energies $> 100$ keV relative to the equatorial boundary of the auroral oval, and how it varies for small and large values of the AE and PC indices of geomagnetic activity. At the end, we shall discuss the role that our results might play on the determination of features of the high-latitude magnetospheric topology.

## 2   Instrumentation and data analysis

We used the data from the METEOR-M1 satellite launched 17 September 2009 into a polar solar-synchronous circular orbit with an altitude of $\approx 830$ km, a period of $\sim 100$ min, and an inclination of $98°$. We used the data of the GGAK-M set of instruments, composed by semiconductor and scintillator detectors, and electrostatic analyzers. In particular, it measured energetic electrons with energies from 0.1 to 13 MeV, and low energy electrons with energies from 0.032 to 16.64 keV (see more details and available data in http://smdc.sinp.msu.ru/index.py?nav=meteor_m1

For automatic detection of the polar boundary of the ORB and the equatorial boundary of the auroral oval we compared the corresponding fluxes with a background reference flux, calculated for each orbit. For energetic particles we calculated the average flux of electrons with energies $> 100$ keV in the polar cap and its standard deviation. We assumed that the measured flux can be classified as ORB electron flux if the difference between this flux and the background flux was greater than five standard deviations during the continuous time interval of at least 1 minute duration (the separate single points spikes are not taken into account). The nearest poleward point that satisfies the described criterion is selected as the polar boundary of the ORB. These selection criteria show stable results of the ORB detection but as a rule they define the boundary at the end of the decline of electron intensity from ORB maximum to the background level. This means that electron fluxes lower than the established criteria, and belonging to the ORB, could be missed. This is why it might shift slightly the obtained boundary equatorward with respect to the true boundary especially in the case of low intensity ORB crossing (see the introduction). This means that we could underestimate the number of events for which the polar boundary of the ORB is observed inside the auroral oval. Such underestimation changes slightly the results of the statistical analysis. However, it cannot change the answer to the main question: whether the trapping boundary is located inside the oval or coincides with its equatorial boundary.

The automatic detection of the polar boundary of ORB, identified as the trapping boundary, might be affected by the sharp local increases in the energetic electron fluxes sometimes observed at the trapping boundary (see Imhof et al., 1990, 1991,

1992, 1993) or just poleward of it. Such fluxes are usually much smaller than the maximum fluxes of the ORB precipitating electrons. Nevertheless, they can be observed during a few hours at the same location in a few consecutive polar satellite orbits (Myagkova et al., 2010; Antonova et al., 2011b; Riazantseva et al., 2012), and alter the automatic detection of the boundary. It was one of the reasons to do a visual inspection of all events.

To calculate the position of the auroral oval boundary, we use the value of the total energy flux. We produce the spectra approximation from 0.032 till 16.64 keV with energy step $d\varepsilon = 0.01$ keV. Energy flux was calculated as the integral characteristic of low energy electron spectrum

$$Flux_\varepsilon = 2\pi \int j(\varepsilon) \cdot \varepsilon d(\varepsilon)$$

(where $j(\varepsilon)$ the flux for current value of energy $\varepsilon$). We first calculated the average value and standard deviation of the electron energy flux measured at $L < 3 \, R_E$, where $L$ is the McIlwain parameter. In the next step we considered the fluxes that exceed the background flux seven standard deviations. If the obtained boundary was located at $L > 3 \, R_E$, we repeated this procedure but calculating the average flux and its standard deviation up to the boundary, determined in the first step. Based on the Vorobjev et al. (2013) definition of the auroral oval, we also imposed additional criterion to the value of the total energy electron flux: it

should be greater than 0.2 erg cm$^{-2}$ s$^{-1}$. The results obtained were also confirmed by a visual inspection.

We used the AE index (Davis and Sugiura, 1966) that represents the dynamics of the auroral electrojet, to identify the intervals of substorm activity. We also used the Polar Cap (PC) index (Troshichev and Andrezen, 1985; Troshichev and Janzhura, 2012), which was created as a proxy of dawn-dusk electric field in the polar cap and Region 1 currents of Iijima and Potemra (1976) intensity. We took for the analysis the one minute values of the AE and PC indices when the spacecraft was at the

equatorial boundary of the auroral oval. Taking into account that there are two PC indices, obtained for the northern (PCN) and southern (PCS) hemispheres, we used the corresponding PCN (PCS) indices for northern (southern) crossings of the auroral oval.

Fig.1 shows an example of two crossings of the auroral oval in the morning and evening MLT sectors on 01 February 2010, when the trapping boundary was located inside the auroral oval. The top panel shows the spectrogram of low energy electrons,

the bottom panel shows total energy flux, calculated from the electron spectra presented on the top (red solid line) and counts of electrons with energy $\geq 100$ keV (green solid line). Dashed red lines in both panels indicate the position of the equatorial boundaries of the auroral oval and dashed green lines show the position of the polar boundaries of ORB. It is clearly seen that the curves of total energy flux and counts of electrons with energy $\geq 100$ keV show the position of the trapping boundary poleward of the equatorial boundary of the auroral oval.

According to the omniweb database (http://omniweb.gsfc.nasa.gov), the solar wind number density ($N_{SW}$) and velocity ($V_{SW}$), and of three components of the interplanetary magnetic field (IMF) for both equatorial borders were very common: $B_x \approx 2$ nT, $B_y \approx -4$ nT, $B_z \approx -1$ nT, $N_{SW} \approx 6$ cm$^{-3}$, and $V_{SW} \approx 450$ km s$^{-1}$. This event took place in the absence of geomagnetic storms ($Dst \approx -7$ nT), and during moderate auroral activity (150 nT $<$AE$<$ 300 nT, and AL$> -300$ nT). The values of PC index were also moderate (PCS$< 3$) (see http://pcindex.org). As it can be seen, for this event the trapping boundary

of energetic electrons, shown by green dashed lines, is located inside the auroral oval. The differences between the latitudes

of the equatorial boundary of the oval and the trapping boundary, $\Delta Lat$ are equal to -5.8° for the dawn and -1.7° for the dusk boundaries.

Fig.2 shows an event of the trapping boundary located outside the auroral oval observed on 17 January 2010. The satellite crossed twice the auroral oval during very quiet geomagnetic conditions ( $B_x \approx 2$ nT, $B_y \approx -1$ nT, $B_z \approx 2.5$ nT, $N_{SW} \approx 6$ cm$^{-3}$, and $V_{SW} \approx 350$ km s$^{-1}$, $Dst \approx -2$ nT, AE$\approx 15$ nT, AL$\approx -15$ nT, PCN< 1. The observed difference was comparatively small: $\Delta Lat = 1°$ for the dawn and 3.3° for the dusk boundaries.

Comparison of events shown in Fig.1 and 2 could bring to a conclusion that the relative location of the trapping boundary and the equatorial boundary of the auroral oval might be affected by the shift of the oval to higher latitudes with the decrease of the geomagnetic activity. However, there are many other events observed for low activity for which the trapping boundary was observed inside the oval. One of examples of such kind of events is shown in Fig.3.

It took place on 26 January 2010 during quiet geomagnetic conditions (IMF $B_x \approx -2.2$ nT, $B_y \approx -4.0$ nT, $B_z \approx -1.5$ nT, $N_{SW} \approx 3.5$ cm$^{-3}$, and $V_{SW} \approx 370$ km s$^{-1}$, $Dst \approx -17$ nT, $AE \approx 50$ nT, $AL \approx -30$ nT, PCS< 1. For this event, $\Delta = -5.1°$ for the dawn and $-2.2°$ for the dusk sectors.Existence of different types of events requires making a statistical analysis to clarify how the geomagnetic conditions could affect the relative location of both boundaries.

## 2.1  Statistical analysis

We analyzed the data from METEOR-M1, obtained for more than 6200 auroral oval and the outer boundary of the ORB crossings. For each crossing, we determined the difference between the geomagnetic latitudes of the equatorial boundary of the auroral oval and of the trapping boundary, $\Delta Lat$. The negative difference $\Delta Lat < 0$ means that the trapping boundary is located inside the auroral oval while the positive difference $\Delta Lat > 0$ indicates that the trapping boundary is located equatorward of the auroral oval. The METEOR-M1 satellite has a sun-synchronous orbit. That is why we obtained $\Delta Lat$ only for a limited range of MLTs.

To analyze how these differences could be affected by geomagnetic activity, we divided all data into two data sets according to the AE or PC indices. Fig.4 shows the distribution of the latitude differences $\Delta Lat$ for AE> 150 nT and AE< 150 nT for the northern (a) and southern (b) hemispheres. As it can be seen, the number of events for which the trapping boundary is observed inside the auroral oval increases significantly with the increase of geomagnetic activity, quantified through the AE index. For AE> 150 nT the trapping boundary is located inside the auroral oval for the majority of events for both hemispheres, while for AE< 150 the trend is not so clear - the number of events where the trapping boundary is located inside and outside of the auroral oval is nearly the same. However, for both sets there are a comparatively large number of events, for which this difference is comparatively small.

Fig.5 shows the distribution of the latitude differences $\Delta Lat$ for PC> 1 and < 1 and for the northern (a) and southern (b) hemispheres, respectively. Comparing Fig.4 and 5, we can see that both distributions are very similar, which can be explained by high correlation between the AE and PC indices obtained by Vennerstrøm et al. (1991). This correlation is related to the formation of ionospheric current systems as a result of the magnetosphere-ionosphere interactions, and the dominant role of the Region 1 currents of Iijima and Potemra (1976) in the formation of the PC index (Troshichev and Janzhura, 2012). However,

the obtained similarity in the behavior of the boundaries, using the AE and PC indices as separate measures of geomagnetic activity, was not evident at the beginning of this study. This supports the picture obtained by Akasofu (1968) in which the trapping boundary is located inside the auroral oval. We underline that the described effect can be clearly seen only in case of simultaneous measurements of plasma and energetic electrons on board of the same satellite, which allow to observe the trapping boundary inside the auroral oval directly during the local measurements. The statistical comparison of boundaries masks this effect, because the scattering of the position of the discussed boundaries in different crossings can be rather large (the standard deviation in the statistical position of the boundaries $\approx \pm 2°$ for the trapping boundaries and $\approx \pm 3°$ for the equatorial boundaries of the auroral oval) whereas the main part of $\Delta Lat$ distributions in Fig.4 and 5 show the difference between boundaries within the limits $\pm 2°$ in case of low geomagnetic activity. The observed scattering in positions of the boundaries are in agreement with early established scattering of the auroral oval boundaries (see Vorobjev et al., 2013, and references therein) and the outer ORB boundary (Kanekal et al., 1998; Kalegaev et al., 2018).

The analysis of the shifts of the studied boundaries with the increase of geomagnetic activity requires special attention and it is far from the main subject of our research. Fig.6 shows the L (McIlwain parameter) – distribution of both boundaries for AE< 150 nT and AE> 150 nT in both hemispheres. It is possible to see the real shift of the equatorial boundary of the auroral oval equatorward with the increase of AE, which is well known due to multiple auroral oval observations. At the same time the position of the trapping boundary practically does not change with the increase of AE. This result is in agreement with Kanekal et al. (1998), in that, in comparison with plasma boundaries, the energetic particle boundaries show a lower degree of correlation with solar wind $B_z$, $V B_z$, and $Kp$ index of geomagnetic activity.

## 3  Discussion and conclusions

We analyzed the relative position of the trapping boundary and the equatorial boundary of the auroral oval using simultaneous measurements of the energetic electrons with energy $> 100$ keV and the auroral electrons made at the same METEOR-M1 satellite. Previous comparisons of the relative position of these boundaries were made mostly statistically using data from different satellites. Our analysis shows that the differences in the positions of both boundaries are typically smaller than the statistical scattering in the position of each boundary. This fact explains why previous statistical studies led to different conclusions, and why the use of statistical results about the location of each boundary cannot answer the question about the relative position of the trapping boundary and the equatorial boundary of the auroral oval.

Our study shows the trapping boundary is often located inside the auroral oval. The number of such events would be enhanced if instruments of better sensitivity were used. This is because the trapping boundary is defined as the boundary where particle fluxes become lower than a threshold determined by the sensitivity of a detector in case of low level of electron flux inside the ORB, so an increase in the sensitivity would move the detected trapping boundary poleward, i.e. deeper inside the auroral oval. The analysis of the latitudinal difference in the position of both boundaries for AE more or less than 150 nT, and for PC more or less than 1 shows that the number of events when the trapping boundary is observed inside the auroral oval significantly increases with both AE and PC indices.

The location of the trapping boundary inside the auroral oval agrees with latest results on the auroral oval mapping discussed by Antonova et al. (2017). They argue that the auroral oval has a form of a comparatively thick ring for all MLTs. Mapping of the plasma sheet to the ionospheric altitudes cannot produce the structure with non-zero thickness near noon. Therefore, it seems natural to map the auroral oval into the plasma ring that surrounds the Earth, as selected by Antonova et al. (2013, 2014a), and filled with plasma similar to the plasma in the plasma sheet. Results of Antonova et al. (2014b, 2015) and Kirpichev et al. (2016) also support such conclusion and locate the quiet time equatorial boundary of the auroral oval at $\sim 7\,R_E$ near midnight and polar boundary at $\sim 10 - 13\,R_E$. It is also important to remember that starting from Vernov et al. (1969) this magnetospheric region is classified as the region of quasitrapping for energetic particles. It contains locked inside the magnetosphere drift trajectories, and only particles with near to $90°$ pitch-angles have drift trajectories crossing the magnetopause. The drift trajectories of particles with other pitch angles are locked inside the magnetosphere. Therefore, the registration of the trapping boundary of energetic electrons with nearly zero pitch angles inside the auroral oval seems quite natural.

The observation of the trapping boundary of energetic electrons inside the oval can also be important for the solution of the problem of acceleration of electrons in the ORB, taking into account that the injection of seed population of relativistic electrons during magnetic storms takes place at the equatorial boundary of the auroral oval (see the results and discussion in Antonova and Stepanova, 2015). Electrons of such seed population must be trapped inside the magnetosphere and further accelerated to relativistic energies during the recovery phase of storm, forming a new ORB. Our current studies were done for comparatively quiet geomagnetic conditions. It also point out the necessity to keep studying the position of the ORB boundaries taking into account an overlapping of the part of the auroral oval and the ORB, using a more sophisticated instrument for the measurement of energetic electrons, and to extend this study to the geomagnetic storm time intervals. For our study we used integrated fluxes of the precipitating electrons with the energy $> 100$ keV. Hence, our results provide the information about an averaged value of polar boundaries, which might vary significantly from the dynamic low energy seed population ($\sim 100$ keV) up to the high (ultra-relativistic energies $> 1$ MeV), taking into account that the seed electron acceleration to higher energies, and the radial diffusion contributes to the redistribution of the electron population (see Reeves et al., 2013; Zhao and Li, 2013; Turner et al., 2015a, b, 2017). It is necessary to add, that the recent results including the observations of the Van Allen Probes has led to significant advances in the study of the dynamics of the ORB. For example, Ripoll et al. (2015) showed the existence of a rather stable core of the ORB. The energy dependence of the inner boundary of the ORB was carefully analyzed by Reeves et al. (2016); Ripoll et al. (2017), and injection of seed population at low latitudes was studied by Turner et al. (2015a). Recent studies (Makarevich et al., 2009; Kunduri et al., 2017; Lejosne and Mozer, 2017) are of a special interest, showing a strong increase of transverse electric fields in subauroral polarization streams (SAPS), which according to Lejosne et al. (2018) can modify the picture of particle injection in the slot region. However, it will be interesting to continue researches of the outer radiation belt considering the results obtained in our paper.

In summary, we can conclude that the trapping boundary of electrons with energy $> 100$ KeV, which coincides with the polar boundary of the ORB, is often located inside the auroral oval. This applies almost always to high geomagnetic activity times and also, though less often, to low geomagnetic activity times. All this that might help to re-analyse the relation between the dynamics of radiation belts and auroral phenomena.

*Data availability.* The data set is available at the experiment METEOR-M1 web page: http://smdc.sinp.msu.ru/index.py?nav=meteor_m1

*Author contributions.* M. O Riazantseva produced the statistical analysis of measurements, E. E.Antonova and M. V. Stepanova were responsible for ideologie, B. V. Marjin and I. A. Rubinshtein designed the instruments, V. O. Barinova produced the primary data processing, N. V. Sotnikov took part in statistical analysis.

5  *Competing interests.* No competing interests are present

*Acknowledgements.* The work of M.S. was supported by Chilean FONDECYT No 1161356 grant, and CONICYT PIA Project Anillo de Investigación en Ciencia y Tecnología ACT1405, and AFORS 695 FA9559-14-1-0139.

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

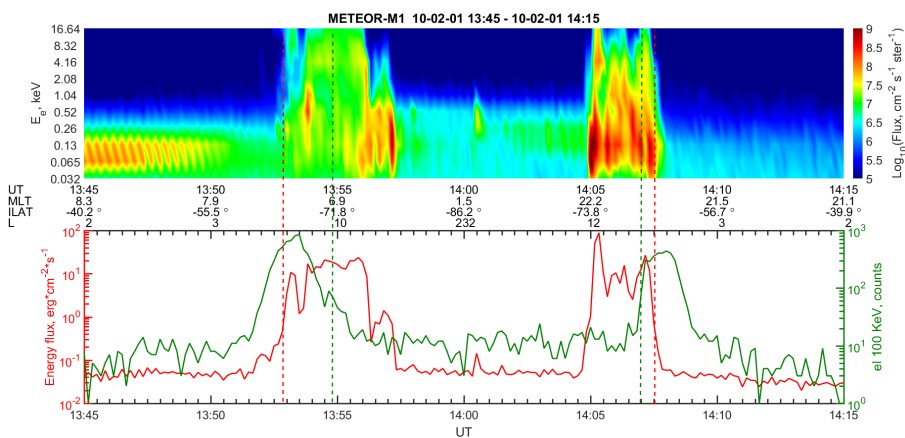

**Figure 1.** An example of the location of the polar boundary of ORB inside the auroral oval at AE> 150 nT. Top panel -spectrogram of low energy electrons, bottom panel: red solid line - total energy flux, calculated from the electron spectra presented on the top; green solid line - counts of electrons with energy $\geq$ 100 KeV; dashed red lines mark the position of the equatorial boundaries of the auroral oval; dashed green lines - the position of the polar boundaries of ORB.

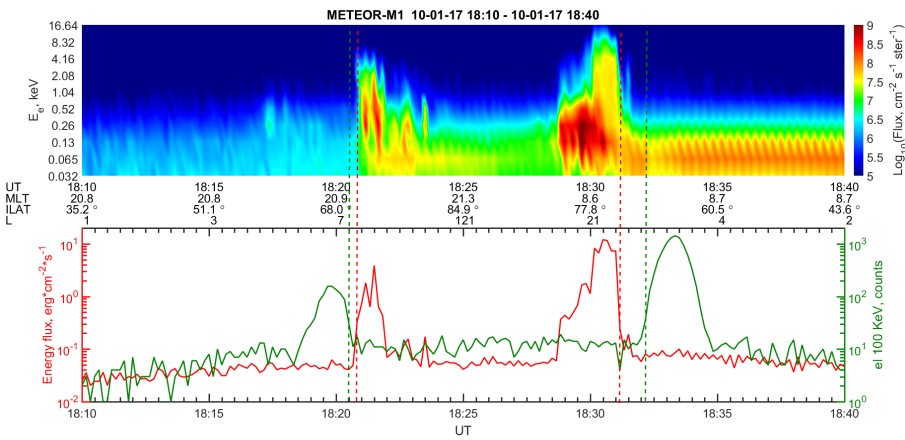

**Figure 2.** An example of observation of the polar boundary of ORB outside the auroral oval at AE< 150 nT. The notations are the same as in Fig.1

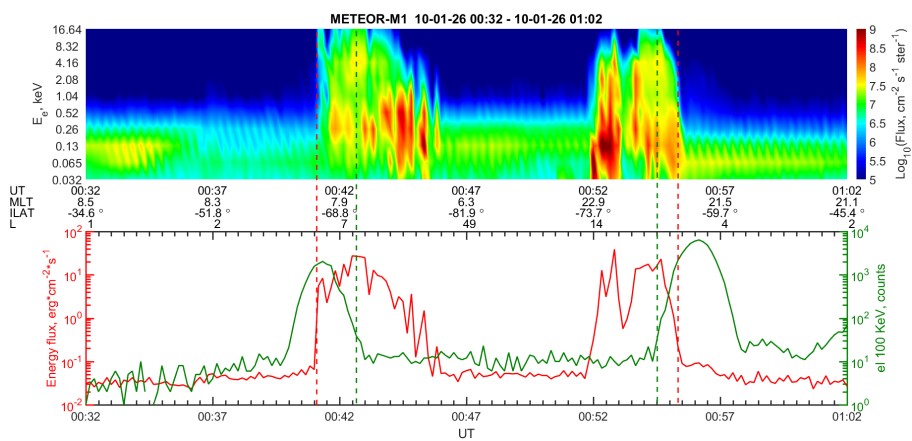

**Figure 3.** An example of observation of polar boundary of ORB inside the auroral oval at AE< 150 nT. The notations are the same as in Fig.1

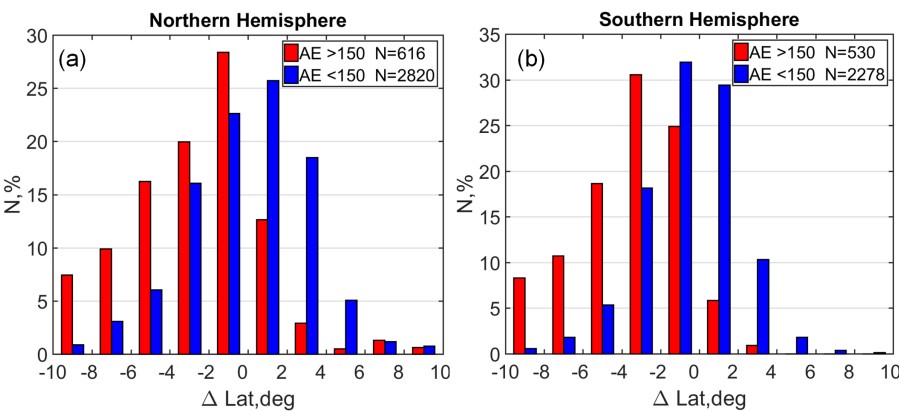

**Figure 4.** The distribution of $\Delta Lat$ for AE>150 nT (red bins) and <150 nT (blue bins) for northern (a) and southern (b) hemispheres. $N$ show the number of events under described criteria.

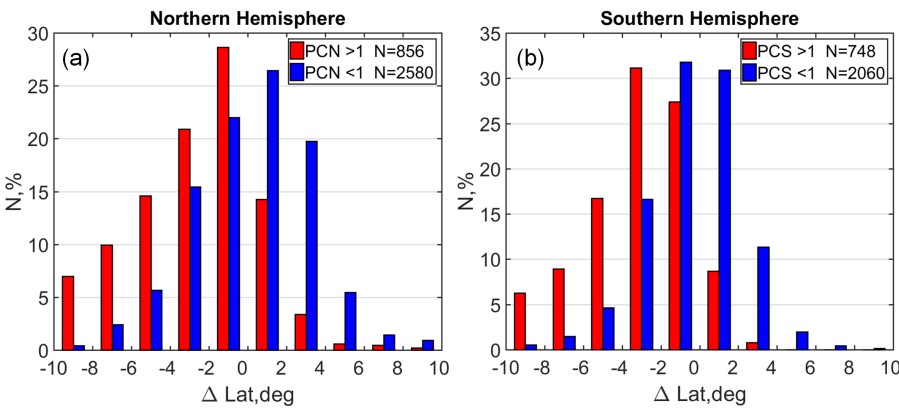

**Figure 5.** The distribution of $\Delta Lat$ for PC>1 (red bins) and <1 (blue bins) for northern (a) and southern (b) hemispheres. $N$ show the number of events under described criteria.

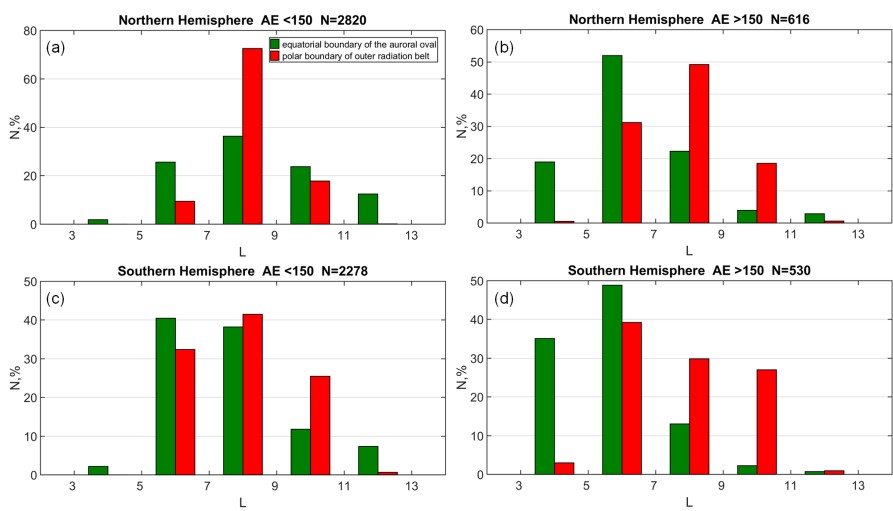

**Figure 6.** The distributions of the position of the equatorial boundary of the auroral oval (green bins) and the polar ORB boundary (red bins) from the $L$ (where $L$ is the McIlwain parameter) for northern (a,b) and southern (c,d) hemispheres for AE< 150 nT (a,c) and AE> 150 nT (b,d).