# Peer review of "Relative positions of the polar boundary of the outer electron radiation belt and the equatorial boundary of the auroral oval"

_Annales Geophysicae, 2018_

## Referee Comment (RC1) · Anonymous Referee #1 · 20 Feb 2018

**General comments**

This paper investigates the location of the external boundary of the outer radiation belt (ORB) relative to the equatorward edge of the auroral oval during quiet or moderately unsettled geomagnetic conditions. The study is based on precipitating electron flux data from the METEOR-M No 1 satellite at auroral (0.03–16 keV) and > 100 keV energies, collected between between November 2009 and March 2010. Three types of situations are exemplified in the paper: (i) external ORB boundary inside the auroral oval during moderately disturbed conditions, (ii) external ORB boundary equatorward from the auroral oval during quiet conditions, and (iii) external ORB boundary inside

the auroral oval during quiet conditions. This gives motivation to carry out a statistical study by looking at the distribution of the separation between the external ORB boundary and the equatorward auroral oval boundary, named d(lat) in the paper, as a function of geomagnetic activity. The distributions are plotted separately for quiet conditions ($AE < 150$ nT or $PC < 1$) and moderately disturbed conditions ($AE > 150$ nT or $PC > 1$). It is found that, during moderate geomagnetic activity, the ORB boundary is located within the auroral oval, whereas during quiet conditions its location can be either inside or outside the auroral oval.

**Specific comments (major)**

The title of the article is somewhat misleading, as it contains the word "relation" which leads one to expect to find an equation (be it empirical) linking the positions of the two studied boundaries. Since no such relation is obtained in the paper, the title should be modified to better reflect the conclusions of the study.

The caption of Figure 1 should be expanded to describe each panel in more detail. It is currently not easy for the reader to understand the data which are plotted, especially what the vertical dashed lines represent. I have not found in the text what the blue and red lines represent, for instance. Moreover, there are many of these lines which seem to be superposed on top of one another, but since the alignment is not perfect, I am not sure whether this is coincidental or done on purpose (same issue with Figure 3). Would it be possible to clarify this and improve the legibility of the figure? Also, it is not so clear why, in the lower panel, the flux energy is plotted, since (if I understood correctly) the criterion for determining the ORB boundary is the $> 100$ keV flux. Unless the blue curve is the integrated version of the fluxes displayed in the top panel? Please clarify this too, since I am not sure whether my guess is correct without additional information in the figure caption (or at the very least in the text describing the figure).

I did not manage to understand the reasoning exposed on p. 3 l. 2–8 (and also mentioned on p. 8 l. 10–14). Why is it so that the energetic electron detector becomes less

sensitive when it is outside of the auroral oval? Since we are here considering a same detector measuring fluxes in one given energy range ($> 100$ keV), why should it not be possible to compare the measurements when they are made inside or outside the auroral oval? To my mind, if such a comparison were not possible to make, this would question the validity of the entire study, since it would be difficult to conclude anything from the data analysis! Could you please explain in more detail or rephrase the idea behind your reasoning in this paragraph?

On p. 8 l. 5–6: "Our analysis shows that the differences in the positions of both boundaries are typically smaller than the statistical scattering in the position of each boundary." I think this statement should be justified with numbers, since currently the "statistical scattering in the position of each boundary" is not quantified in the paper. This should be easy to add, as you already have made a statistical study of the boundary locations, and there are certainly many references in the literature that could be cited to support the said statement.

The conclusions presented on p. 9 ("there [is] strong evidence that [the] trapping boundary of energetic electrons [...] is located inside the auroral oval") do not reflect the interpretation of Figures 4 and 5. One cannot neglect the relatively high number of events for which this trapping boundary is situated equatorwards from the auroral oval, so the quoted statement is misleading.

Finally, I think it could be extremely interesting to go a bit further in the analysis before the final publication of the manuscript, by trying to determine why d(lat) changes with increasing geomagnetic activity (from totally quiet to moderate activity). Is it so that only the auroral oval equatorward boundary moves equatorwards, while the ORB external boundary does not change, or does the ORB boundary also migrate equatorwards/polewards when geomagnetic activity is enhanced? If such a result could be obtained, this would to my mind greatly increase the impact of the paper, and this would enable one to deepen the interpretation of the results.

**Specific comments (minor)**

– The acronym "ORB", which first appears on p. 2 l. 24 (and most probably stands for "outer radiation belt") should be defined in the introduction.

– p. 2 l. 28: "After that we searched for the closest to the pole location of the ORB flux" does not sound very clear to the reader. This should be rephrased.

– p. 3 l. 14: I would suggest to add the reference to Davis and Sugiura (1966) on the AE index, since references are provided for the PC indices.
Davis, T. N., and M. Sugiura (1966), Auroral electrojet activity index AE and its universal time variations, J. Geophys. Res., 71, 785–801, doi:10.1029/JZ071i003p00785.

– p. 4 l. 22–23: "According to the (http://omniweb.gsfc.nasa.gov/)..." → There must be several words missing here!

– p. 7: Could you explain in a little more detail why you chose the value of 150 nT for the AE index to separate the events in the analysis? What would happen if you chose, say, AE = 100 nT instead? Would the trend for low geomagnetic activity become clearer? (cf l. 6)

– p. 7 l. 14–15: "using the AE and PC ind[ices] as a measure of geomagnetic activity by separately" –> there must be words missing here too!

**Technical corrections**

– "indexes" → "indices" (p. 1 l. 22; p. 3 l. 13–16; p. 4 l. 18–19; p. 7 l. 2–11–14)

– p. 1 l. 16: "at the absence of" → "**in** the absence of"

– p. 1 l. 18–19: "to the equator from" → "equatorward from" (same p. 2 l. 3)

– p. 1 l. 19, l. 22: "auroral precipitations" → "auroral precipitation" ("precipitation" is uncountable)

– p. 1 l. 24: "is discussed" → "are discussed"

– p. 1 l. 25: "the position of the trapping boundary **for** energetic electrons"

– p. 1 l. 26: "sing" → "using"

– p. 1 l. 26: "low orbiting and high apogee" → "low-orbiting and high-apogee" (same l. 28, p. 2 l. 4)

– p. 2 l. 32: remove comma after "it is well known"

– p. 3 l. 9: "location" → "locations" (or change "have" into "has" on l. 11; same l. 11)

– p. 3 l. 17: "high latitude" → "high-latitude"

– p. 3 l. 20: "of GGAK-M set" → "of **the** GGAK-M set"

– p. 3 l. 22: "with the energies from..." → "with energies from..." (twice on this line)

– p. 3 l. 29: "as a polar boundary" → "as **the** polar boundary"

– p. 4 l. 2–3: correct the location of parentheses for the citations

– p. 4 l. 6: "the visual inspection" → "**a** visual inspection"

– p. 4 l. 18–19: remove capitalisation of "Northern" and "Southern" (see guidelines: https://www.annales-geophysicae.net/for_authors/manuscript_preparation.html)

– p. 6 l. 13: "trapping boundary d(lat)" → "trapping boundary, d(lat)" (add comma)

– p. 7 l. 14: "behaviour" → "behavior" (to remain consistent with p. 9 l. 1 and the use of American English spelling throughout the paper)

– p. 7 l. 16: I think "1.2 Subsection (as Heading 2)." should be deleted.

– p. 8 l. 5: "using the data from" → "using data from"

– p. 8 l. 23: "quite time" → "quiet time"

– p. 8 l. 27: "with another pitch angles" → "with other pitch angles"

– p. 8 l. 29: "can be also" → "can also be"

– p. 9 l. 3: "there are strong evidences" → "there is strong evidence" ("evidence" is uncountable)

– p. 9 l. 3: "that trapping boundary" → "that **the** trapping boundary"

---

## Referee Comment (RC2) · Anonymous Referee #2 · 8 Mar 2018

This paper presents potentially interesting results and interpretations. With a little more detail within the manuscript, and slightly more interaction between the introduction and the conclusions sections it will provide a useful scientific step forward.

Some comments regarding the text and figures are presented below:

1) In the paragraph starting page 2, Line 19 two mechanisms are put forward for the relative locations of the equatorial boundary of the auroral oval and the outer radiation belt trapping boundary. The rest of the paper is about determining which mechanism is supported by the analysis of satellite data as presented. However, the opening sentence of page 8, line 18 indicates that the results agree with Anotonova et al. 2017.

[Figure]

This work was not mentioned in the Introduction section and therefore is not expected. The new work should be discussed in section 1 to give the reader the background to the research mentioned in that paper.

2) The first paragraph of section 1 discusses the L-shell variations of the boundaries, particularly the outer radiation belt trapping boundary. Given the use of 100 keV in this study to determine the boundary location rather than 40 keV or 35 keV as previously used, it would be beneficial to the paper if the distributions in L-shell of the boundaries were plotted for the whole dataset - similar to Figures 4 and 5. These new figure(s) would provide clarity for the reader and confirm that the algorithm is producing results that are consistent with the previous work cited in paragraph 1&2, section 1.

3) Figures 4 and 5 show the distributions of the boundaries for northern and southern hemispheres. However, no obvious follow-up of this separation is undertaken, and it is unclear why it is done. It is reasonable to use the PCS index for the southern hemisphere analysis, but it is unclear why the data continue to be separated hemispherically after that. Just having one plot for each activity index would clarify the presentation and aid the discussion of the main result, i.e., that there is a latitudinal difference in the distributions for quiet and active conditions.

Some small points:

4) 'to the equator of' should be replaced by 'equatorward of'. 'to the pole of' should be replaced by 'poleward of'.

5) Page 2, line 4-5. The sentence is unclear. I think it says that the outer radiation belt trapping boundary is clearly identifiable in low orbiting satellite data.

6) It would be useful to the reader to state whether the electron detector was measuring spin averaged electrons or was omni-directional etc.

7) Page 4, line 7-8. What energy did you use to calculate the average value and std of the electron fluxes? Same question for the total energy electron flux. If all of the

auroral electron energy data in the range from 0.032-16.64 keV was used, how was it combined?

8) Figure 1. The caption should describe the lines added to the plot. What does the red vertical dashed line represent. The caption should say - the text doesn't. Why are there two green vertical lines at ∼14:06 UT. Why is there a red vertical line in Figure 1 and a blue vertical line in Figure 2?

---

## Author Comment (AC1) · 11 Apr 2018

**General comments**

This paper investigates the location of the external boundary of the outer radiation belt (ORB) relative to the equatorward edge of the auroral oval during quiet or moderately unsettled geomagnetic conditions. The study is based on precipitating electron flux data from the METEOR-M No 1 satellite at auroral (0.03–16 keV) and > 100 keV energies, collected between between November 2009 and March 2010. Three types of situations are exemplified in the paper: (i) external ORB boundary inside the auroral oval during moderately disturbed conditions, (ii) external ORB boundary equatorward

from the auroral oval during quiet conditions, and (iii) external ORB boundary inside the auroral oval during quiet conditions. This gives motivation to carry out a statistical study by looking at the distribution of the separation between the external ORB boundary and the equatorward auroral oval boundary, named d(lat) in the paper, as a function of geomagnetic activity. The distributions are plotted separately for quiet conditions (AE < 150 nT or PC < 1) and moderately disturbed conditions (AE > 150 nT or PC > 1). It is found that, during moderate geomagnetic activity, the ORB boundary is located within the auroral oval, whereas during quiet conditions its location can be either inside or outside the auroral oval.

We are grateful for the great work done by you with our article and for the list of useful comments and corrections! We hope that the new version of the paper become better and more understandable for readers.

1. The title of the article is somewhat misleading, as it contains the word "relation" which leads one to expect to find an equation (be it empirical) linking the positions of the two studied boundaries. Since no such relation is obtained in the paper, the title should be modified to better reflect the conclusions of the study.

Thank you, the new title is: "Relative locations of the polar boundary of outer electron radiation belt and the equatorial boundary of the auroral oval"

2.The caption of Figure 1 should be expanded to describe each panel in more detail. It is currently not easy for the reader to understand the data which are plotted, especially what the vertical dashed lines represent. I have not found in the text what the blue and red lines represent, for instance. Moreover, there are many of these lines which seem to be superposed on top of one another, but since the alignment is not perfect, I am not sure whether this is coincidental or done on purpose (same issue with Figure 3). Would it be possible to clarify this and improve the legibility of the figure? Also, it is not so clear why, in the lower panel, the flux energy is plotted, since (if I understood correctly) the criterion for determining the ORB boundary is the > 100 keV flux. Unless the blue curve

is the integrated version of the fluxes displayed in the top panel? Please clarify this too, since I am not sure whether my guess is correct without additional information in the figure caption (or at the very least in the text describing the figure).

We corrected the figures 1-3, trying to make them clearer and added the corresponding notation for all the curves. Also we added some additional comments to the text, see p. 4 l. 27-32

3.I did not manage to understand the reasoning exposed on p. 3 l. 2–8 (and also mentioned on p. 8 l. 10–14). Why is it so that the energetic electron detector becomes less sensitive when it is outside of the auroral oval? Since we are here considering a same detector measuring fluxes in one given energy range (> 100 keV), why should it not be possible to compare the measurements when they are made inside or outside the auroral oval? To my mind, if such a comparison were not possible to make, this would question the validity of the entire study, since it would be difficult to conclude anything from the data analysis! Could you please explain in more detail or rephrase the idea behind your reasoning in this paragraph?

Thank you for the comment! We did not explain our idea sufficiently accurately in the text, which is now is corrected. The sensitivity of the detector is naturally fixed, and does not depend on the location and time of the measurements. We mean the well-known effect of decreasing of the electron fluxes inside the ORB with decreasing level of geomagnetic activity; for example during the periods of minimum solar activity (see, for example, McIlwain C.E., Processes Acting Upon Outer Zone Electrons, Radiation Belts: Model and Standard, Geophysical Monograph, pp. 15-26, 1996.). The observations presented were obtained during such period (September 2009 - April 2010) and sometimes the electron flux in the ORB were very weak, close to the sensitivity limit of the detector. In these cases, we can only detect the beginning of the decline from the ORB maximum to the background level of the electron intensity. In such situations, the detected boundary can be shifted to the equator relative to the true boundary of this low intensity ORB, which could be observed by a detector with better sensitivity. That's

why we believe that the discussed effects could be clearer in the period of solar maximum activity or if the sensitivity of the detector was better. We added some additional comments on p. 3 l.1-4 and l. 24-31

4.On p. 8 l. 5–6: "Our analysis shows that the differences in the positions of both boundaries are typically smaller than the statistical scattering in the position of each boundary." I think this statement should be justified with numbers, since currently the "statistical scattering in the position of each boundary" is not quantified in the paper. This should be easy to add, as you already have made a statistical study of the boundary locations, and there are certainly many references in the literature that could be cited to support the said statement.

Thank you for the comment! We added some additional comments and statistical numbers at the end of the section 3 (p.8. l. 13-20 p.9 l.1-2) with corresponding references.

5. The conclusions presented on p. 9 ("there [is] strong evidence that [the] trapping boundary of energetic electrons [...] is located inside the auroral oval") do not reflect the interpretation of Figures 4 and 5. One cannot neglect the relatively high number of events for which this trapping boundary is situated equatorwards from the auroral oval, so the quoted statement is misleading.

Thank you for the comment! You are right this statement is too categorical. We have corrected it and aligned with the discussed results (see p.10. l 31-33)

6.Finally, I think it could be extremely interesting to go a bit further in the analysis before the final publication of the manuscript, by trying to determine why d(lat) changes with increasing geomagnetic activity (from totally quiet to moderate activity). Is it so that only the auroral oval equatorward boundary moves equatorwards, while the ORB external boundary does not change, or does the ORB boundary also migrate equatorwards/polewards when geomagnetic activity is enhanced? If such a result could be obtained, this would to my mind greatly increase the impact of the paper, and this

would enable one to deepen the interpretation of the results.

Thank you for the comment! The increasing of geomagnetic activity affects first of all the position of the equator boundary of the auroral oval (see, for example, Feldstein et al. (2014, doi: doi:10.5194/hgss-5-81-2014). The position of the polar ORB boundary is more stable (see Kanekal et al. (1998)). The figures 1.1 below show the distributions of the position of both boundaries by Meteor-M1 measurements in McIlwain coordinates (separately for Northern Hemisphere, Sothern Hemisphere, for AE<150 nT and AE>150 nT). The distributions are rather wide, but you can clearly see that the maximum of distributions for polar boundary of ORB is rather stable and don't show any clear dependence on geomagnetic activity. On the other hand the maximum of distributions of equator boundary of auroral oval clearly moves toward the equator with increasing geomagnetic activity. Nevertheless, this is not a simple question because the distributions are rather wide and their widths increase with enhanced geomagnetic activity (for both boundaries). This means that the boundaries position (including polar ORB boundary) are unstable in these cases, and we cannot unequivocally confirm that the polar ORB boundary does not depend on geomagnetic activity. This question needs more thorough study and we don't want to add this discussion to the paper. The main aim of this paper is to show that the polar ORB boundary can be observed rather often inside the auroral oval. It is a very important point for the problem of the ORB formation. So, we introduce new figure (fig.6) and text in the paper with the discussion of the dependence of studied boundaries on geomagnetic activity (section 3 p.9 l. 3-9).

Specific comments (minor)

– The acronym "ORB", which first appears on p. 2 l. 24 (and most probably stands for "outer radiation belt") should be defined in the introduction.

Thank you for the comment! We defined the acronym ORB in the Introduction (p.1 l.24)

– p. 2 l. 28: "After that we searched for the closest to the pole location of the ORB flux" does not sound very clear to the reader. This should be rephrased.

[Figure]

Thank you! We have tried to make this sentence clearer. (P.2 l. 2-3)

– p. 3 l. 14: I would suggest to add the reference to Davis and Sugiura (1966) on the AE index, since references are provided for the PC indices. Davis, T. N., and M. Sugiura (1966), Auroral electrojet activity index AE and its universal time variations, J. Geophys. Res., 71, 785–801, doi:10.1029/JZ071i003p00785.

Thank you for the reference! We have added it at p.4 l. 19.

– p. 4 l. 22–23: "According to the (http://omniweb.gsfc.nasa.gov/)..." → There must be several words missing here!

Thank you for the comment! We mean "According to the omniweb database....". We corrected the corresponding phrase (p.5 l.1) .

– p. 7: Could you explain in a little more detail why you chose the value of 150 nT for the AE index to separate the events in the analysis? What would happen if you chose, say, AE = 100 nT instead? Would the trend for low geomagnetic activity become clearer? (cf l. 6)

Thank you for the comment! Unfortunately, geomagnetic activity was rather low during the observed period (November 2009 - March 2010), so we can't use traditional criteria for disturbed periods. AE 150 nT was selected as a compromise between the idea of separation of disturbed and quiet periods, and the volume of the statistic. If we change the selection criteria to AE = 100 nT, the results do not change significantly (see the figure 1.2 for AE>150 nT, AE<150 nT (a,b), and below for AE>100 nT, AE<100 nT (c,d) ). If we changed the selection criteria significantly to make a strong difference between the geomagnetic conditions (for example to select AE>500 nT and AE<10nT (see the panel (e,f) on the figure 1.2)) we can see that the trapping boundary would always be located inside the auroral oval for AE>500 nT, but the statistic of such crossings is rather poor for the observed period.

– p. 7 l. 14–15: "using the AE and PC ind[ices] as a measure of geomagnetic activity

by separately" –> there must be words missing here too

Thank you! I have changed slightly this sentence (p.8 l.11-12)

– "indexes" → "indices" (p. 1 l. 22; p. 3 l. 13–16; p. 4 l. 18–19; p. 7 l. 2–11–14)

– p. 1 l. 16: "at the absence of" → "in the absence of"

– p. 1 l. 18–19: "to the equator from" → "equatorward from" (same p. 2 l. 3) C4 ANGEOD Interactive comment Printer-friendly version Discussion paper

– p. 1 l. 19, l. 22: "auroral precipitations" → "auroral precipitation" ("precipitation" is uncountable)

– p. 1 l. 24: "is discussed" → "are discussed"

– p. 1 l. 25: "the position of the trapping boundary for energetic electrons"

– p. 1 l. 26: "sing" → "using"

– p. 1 l. 26: "low orbiting and high apogee" → "low-orbiting and high-apogee" (same l. 28, p. 2 l. 4)

– p. 2 l. 32: remove comma after "it is well known"

– p. 3 l. 9: "location" → "locations" (or change "have" into "has" on l. 11; same l. 11)

– p. 3 l. 17: "high latitude" → "high-latitude"

– p. 3 l. 20: "of GGAK-M set" → "of the GGAK-M set"

– p. 3 l. 22: "with the energies from..." → "with energies from..." (twice on this line)

– p. 3 l. 29: "as a polar boundary" → "as the polar boundary"

– p. 4 l. 2–3: correct the location of parentheses for the citations

– p. 4 l. 6: "the visual inspection" → "a visual inspection"

– p. 4 l. 18–19: remove capitalisation of "Northern" and "Southern" (see guidelines: https://www.annales-geophysicae.net/for$_authors/manuscript_preparation.html$)

– p. 6 l. 13: "trapping boundary d(lat)" → "trapping boundary, d(lat)" (add comma)

– p. 7 l. 14: "behaviour" → "behavior" (to remain consistent with p. 9 l. 1 and the use of American English spelling throughout the paper)

– p. 7 l. 16: I think "1.2 Subsection (as Heading 2)." should be deleted.

– p. 8 l. 5: "using the data from" → "using data from"

– p. 8 l. 23: "quite time" → "quiet time"

– p. 8 l. 27: "with another pitch angles" → "with other pitch angles"

– p. 8 l. 29: "can be also" → "can also be"

– p. 9 l. 3: "there are strong evidences" → "there is strong evidence" ("evidence" is uncountable)

– p. 9 l. 3: "that trapping boundary" → "that the trapping boundary"

Thank you for careful reading of our paper! The text was corrected according to your comments and corrections!

Please also note the supplement to this comment:
https://www.ann-geophys-discuss.net/angeo-2018-6/angeo-2018-6-AC1-supplement.zip

———————————————————

[Figure]

[Figure]

**Figure 1.1**: The distributions of the position of equatorial boundary of the auroral oval (green bins) and the polar ORB boundary (red bins) from the L (where L is the McIlwain parameter) for northern (a,b) and southern (c,d) hemispheres for AE <150 nT (a,c) and AE>150 nT (b,d).

**Fig. 1.**

a)

b)

c)

d)

e)

f)

**Figure 1.2:** The distribution of Δ Lat for AE>150 nT and <150 nT (a,b)
for AE>100 nT and <100 nT (c,d) and for AE>500 nT and <10 nT (e,f)
for northern (a,c,e) and southern (b,d,f) hemispheres

**Fig. 2.**

Answer to Referee 1:

*Anonymous Referee #1*

*General comments*

*This paper investigates the location of the external boundary of the outer radiation belt (ORB) relative to the equatorward edge of the auroral oval during quiet or moderately unsettled geomagnetic conditions. The study is based on precipitating electron flux data from the METEOR-M No 1 satellite at auroral (0.03–16 keV) and > 100 keV energies, collected between November 2009 and March 2010. Three types of situations are exemplified in the paper: (i) external ORB boundary inside the auroral oval during moderately disturbed conditions, (ii) external ORB boundary equatorward from the auroral oval during quiet conditions, and (iii) external ORB boundary inside the auroral oval during quiet conditions. This gives motivation to carry out a statistical study by looking at the distribution of the separation between the external ORB boundary and the equatorward auroral oval boundary, named d(lat) in the paper, as a function of geomagnetic activity. The distributions are plotted separately for quiet conditions (AE < 150 nT or PC < 1) and moderately disturbed conditions (AE > 150 nT or PC > 1). It is found that, during moderate geomagnetic activity, the ORB boundary is located within the auroral oval, whereas during quiet conditions its location can be either inside or outside the auroral oval.*

We are grateful for the great work done by you with our article and for the list of useful comments and corrections! We hope that the new version of the paper become better and more understandable for readers.

*1. The title of the article is somewhat misleading, as it contains the word "relation" which leads one to expect to find an equation (be it empirical) linking the positions of the two studied boundaries. Since no such relation is obtained in the paper, the title should be modified to better reflect the conclusions of the study.*

Thank you, the new title is: "Relative locations of the polar boundary of outer electron radiation belt and the equatorial boundary of the auroral oval"

*2. The caption of Figure 1 should be expanded to describe each panel in more detail. It is currently not easy for the reader to understand the data which are plotted, especially what the vertical dashed lines represent. I have not found in the text what the blue and red lines represent, for instance. Moreover, there are many of these lines which seem to be superposed on top of one another, but since the alignment is not perfect, I am not sure whether this is coincidental or done on purpose (same issue with Figure 3). Would it be possible to clarify this and improve the legibility of the figure? Also, it is not so clear why, in the lower panel, the flux energy is plotted, since (if I understood correctly) the criterion for determining the ORB boundary is the > 100 keV flux. Unless the blue curve is the integrated version of the fluxes displayed in the top panel? Please clarify this too, since I am not sure whether my guess is correct without additional information in the figure caption (or at the very least in the text describing the figure).*

We corrected the figures 1-3, trying to make them clearer and added the corresponding  notation for all the curves.  Also we added some additional comments to the text, see p. 4  l. 27-32

*3. I did not manage to understand the reasoning exposed on p. 3 l. 2–8 (and also mentioned on p. 8 l. 10–14). Why is it so that the energetic electron detector becomes less sensitive when it is outside of the auroral oval? Since we are here considering a same detector measuring fluxes in one given energy range (> 100 keV), why should it not be possible to compare the measurements when they are made inside or outside the auroral oval? To my mind, if such a comparison were not possible to make, this would question the validity of the entire study, since it would be difficult to conclude anything from the*

**Fig. 3.**

---

## Author Comment (AC3) · 11 Apr 2018

This paper presents potentially interesting results and interpretations. With a little more detail within the manuscript, and slightly more interaction between the introduction and the conclusions sections it will provide a useful scientific step forward. Some comments regarding the text and figures are presented below:

We are grateful for your careful reading of our paper! We try to improve the paper taking into account all your comments.

1) In the paragraph starting page 2, Line 19 two mechanisms are put forward for the

relative locations of the equatorial boundary of the auroral oval and the outer radiation belt trapping boundary. The rest of the paper is about determining which mechanism is supported by the analysis of satellite data as presented. However, the opening sentence of page 8, line 18 indicates that the results agree with Anotonova et al. 2017. This work was not mentioned in the Introduction section and therefore is not expected. The new work should be discussed in section 1 to give the reader the background to the research mentioned in that paper.

The main idea presented in this study rose once it became clear that the main part of the auroral oval is not mapped onto the plasma sheet, as it used to be widely accepted. According to our previous studies, the oval is mapped onto the surrounding-the-Earth plasma ring. The existence of such ring, which exhibits characteristics similar to the plasma sheet, was known from the first satellite plasma measurements (see, for example, (Frank, 1971, doi:10.1029/JA076i010p02265). Transverse currents in this ring are closed inside the magnetosphere. So we added a discussion of the results by Anotonova et al.(2017) in the Introduction p.2 .l.24-25

2) The first paragraph of section 1 discusses the L-shell variations of the boundaries, particularly the outer radiation belt trapping boundary. Given the use of 100 keV in this study to determine the boundary location rather than 40 keV or 35 keV as previously used, it would be beneficial to the paper if the distributions in L-shell of the boundaries were plotted for the whole dataset - similar to Figures 4 and 5. These new figure(s) would provide clarity for the reader and confirm that the algorithm is producing results that are consistent with the previous work cited in paragraph 12, section 1.

Thank you for your comment! We use the lowest channel of energetic electrons (>100 keV) available on Meteor-M1 satellite to determine the trapping boundary. Below we show plots (figure 2.1) of the probability distributions of finding the obtained boundaries for each L value (where L is McIlwain parameter). For quiet geomagnetic condition the average value of the polar boundary of the ORB ≈ 8±1, the average value of the equatorial auroral oval boundary is almost the same ≈ 8±2. For perturbed geomagnetic

condition AE>150 nT the average value of the polar boundary of the ORB is also ≈ 8±1, whereas the average value of the equatorial auroral oval boundary is much less ≈ 6±2 . The average position of the polar boundary of the ORB agrees with the position of the trapping boundary published by Vernov et al. (2009). We added figures and comments in section 3 p.9. l.3-9

3) Figures 4 and 5 show the distributions of the boundaries for northern and southern hemispheres. However, no obvious follow-up of this separation is undertaken, and it is unclear why it is done. It is reasonable to use the PCS index for the southern hemisphere analysis, but it is unclear why the data continue to be separated hemispherically after that. Just having one plot for each activity index would clarify the presentation and aid the discussion of the main result, i.e., that there is a latitudinal difference in the distributions for quiet and active conditions.

The maximums of distributions of $\Delta$Lat for northern and southern hemispheres (see the figure 2.2 below) are slightly different (in adjacent bins). The combined distribution, for both hemispheres, exhibits a smeared maximum, and the effect is less clear. During the data analysis we found important differences using the AE and the PC indexes. The AE index is produced only due to magnetic measurements in the northern hemisphere. At the same time, the PC index exists separately for northern and southern hemispheres. We obtain slightly different pictures of $\Delta$Lat for both hemispheres. We do not know whether this effect is connected to the difference in magnetic field between both hemispheres (IGRF effect) or some kind of seasonal effect (our measurements were made on September 2009 - April 2010). It could be very interesting to clarify this subject in the future. This is why we prefer to publish our figures without averaging both hemispheres.

Some small points:

4) 'to the equator of' should be replaced by 'equatorward of'. 'to the pole of' should be replaced by 'poleward of'.

Thank you! We have corrected the terms everywhere.

5) Page 2, line 4-5. The sentence is unclear. I think it says that the outer radiation belt trapping boundary is clearly identifiable in low orbiting satellite data.

Thank you! We have corrected this sentence. ( p. 2 l.2-3.)

6) It would be useful to the reader to state whether the electron detector was measuring spin averaged electrons or was omni-directional etc.

Unfortunately, we have no information on the pitch-angle distribution of both auroral electrons and energetic electrons. METEOR-1 satellites were spin stabilized. The detectors of GGAK-M instrument look within the loss cone and mostly observe precipitating particles. However, because of the large fields of view and particle scattering, some amount of the trapped population is also seen. We used early published information about the isotropy of the observed fluxes of energetic electrons near the ORB from Imhof et al. (1990, 1991, 1992, 1993). Auroral oval was identified by precipitating low energy electrons.

7) Page 4, line 7-8. What energy did you use to calculate the average value and std of the electron fluxes? Same question for the total energy electron flux. If all of the auroral electron energy data in the range from 0.032-16.64 keV was used, how was it combined?

The polar boundary of ORB was determined using the average flux of electrons with energy >100 keV. The equator boundary of the auroral oval was determined using the value of the total energy flux of low energy electrons. Each spectra was approximated in the range 0.032-16.64 keV with an energy step $d\epsilon$ =0.01 keV, and the energy flux was calculated as a numerical integral

$$Flux_\epsilon = 2\pi \int (j(\epsilon) \cdot \epsilon \, d\epsilon \qquad (1)$$

( $j(\epsilon)$ - flux for current value of energy $\epsilon$ ). We have added the corresponding explanations in the text (p. 4 l.10-13).

8) Figure 1. The caption should describe the lines added to the plot. What does the red vertical dashed line represent. The caption should say - the text doesn't. Why are there two green vertical lines at 14:06 UT. Why is there a red vertical line in Figure 1 and a blue vertical line in Figure 2?

We are sorry. I. The new version contains corrected and improved figures 1-3.

Please also note the supplement to this comment:
https://www.ann-geophys-discuss.net/angeo-2018-6/angeo-2018-6-AC3-supplement.zip

———————————————————

[Figure]

**Figure 2.1:** The distributions of the position of equatorial boundary of the auroral oval (green bins) and the polar ORB boundary (red bins) from the L (where L is the McIlwain parameter) for northern (a,b) and southern (c,d) hemispheres for AE <150 nT (a,c) and AE>150 nT (b,d).

**Fig. 1.**

[Figure]

**Figure 2.2:** The distribution of Δ Lat for AE>150 nT (red bins) and <150 nT (blue bins) for northern (a) and southern (b) and combined northern and southern (c) hemispheres

**Fig. 2.**

---

## Referee Report (RR1)

**Re-review of paper #angeo-2018-6 – "Relative locations of the polar boundary of the outer electron radiation belt and the equatorial boundary of the auroral oval" by Riazanteseva et al.**

**Overall comment**

In this revised manuscript, the authors have addressed all my comments in a satisfactory way. To my mind, the paper is now clear and reaches substantial conclusions of interest for the scientific community. I therefore recommend the paper for publication, provided the few technical corrections given below are made.

**Copyediting and typesetting**

– p. 1, l. 27: high-apogee

– p. 4, l. 1: whether the trapping boundary

– p. 8, l. 11: as separate measures

– p. 10, l. 9: so the increasing of --> so an increase in

– p. 11, l. 2: Investigacin --> Investigación; Tecnologa --> Tecnología

---

## Referee Report (RR2)

Dear Editor,

This is my review of the paper « Relative locations of the polar boundary of the outer electron radiation belt and the equatorial boundary of the auroral oval » by M. O. Riazanteseva et al. submitted to AnGeo.

This article addresses the problem of finding the position of the polar boundary of the outer electron radiation belt (ORB), relative to the position of the auroral oval (AO). The authors perform a statistical study using the data of the METEOR-M No1 auroral satellite for the period from 11 November 2009 to 27 March 2010. From it they deliver the respective position of the two structures.

I am the third reviewer of this article and I do recommend publications based on the following comments (A) and once the few additional changes I am asking (B) below are made.

A)      As a third reviewer, I can see how this article has been improved since its submission. I agree with Reviewer 2 that this article provides a useful scientific step forward (now that some modifications have been made. Rewiewer 1 asked to go a bit further in the analysis by trying to determine why d(lat) changes with increasing geomagnetic activity (from totally quiet to moderate activity). This has been done by the authors through Figure 4 and 5. Figure 4 showing the penetration of the ORB into the AO in terms of delta_Latitude is important. Reviewer 2 asked « new figure(s) that would provide clarity for the reader and confirm that the algorithm is producing results that are consistent with the previous work cited in paragraph 1&2, section 1 ». This has been done; figure 6 showing the distributions of the position of equatorial boundary of the auroral oval (green bins) and the polar ORB boundary (red bins) from the L different AE index is important. The authors have therefore accounted for the changes asked by both reviewers and their final figures are 1- clear to me, 2-bringing a statistical description for different geomagnetic activities, 3- support well the conclusions of the article.

B)      There were issues on the energy range for the determination of the ORB and AO. In my review below, I come back on the energy-dependence that is a key point to discuss. The main modification I am asking should be fast to do as I only ask 1- to include a small discussion on the energy-dependence of the ORB (with a link to SAPS), 2- to account for recent publications.

My intension is to recommend publication in AnGeo and to see this article published within short delays.

**Main corrections:**

**1- : On the energy-dependence**

Although authors focus on the polar boundary of the outer belt, I ask here the question of the impact of energy dependence of the outer belt on its shape, and,

therefore, on its boundaries. The inner boundary of the outer belt is extremely energy dependent as shown very recently (e.g. Reeves et al. 2016; Ripoll et al. JGR 2017 and references in them) from the Van Allen Probes, from L~4 to 6. As L-increases above L~6 reaching now the polar boundary of the outer belts, in which the authors are working on, some of this energy-dependence will be kept, in particular for quiet times, for which the plasmasphere extends up to there. From Figure 6 of the authors, it is interesting to see that the probability is higher in the Southern hemisphere. (As a comment, the position of the plasmasphere is determinant on the energy-dependence of the ORB structure, but not only, as wave-particle interactions (WPI) from Chorus waves outside of the plasmasphere will also print an energy-dependence structure of the outer belt). What is interesting about quiet times, on which the authors focus on (but not only), is the great stability of the outer belt. For instance Ripoll et al. 2014 analyzing HEO data shown months of stability of the outer belt for E>100 keV, reinforcing the relevance of defining outer belt boundaries and positioning them onto known structures (aurora oval, ring plasma or current, plasma sheet, etc.). Still, the stability degrades as L-shell increases above 5-6. (As a comment, there must be more recent observations from the Van Allen Probes, which we let the authors find themselves). During, active times, because WPI are extremely energy-dependent, it is likely that the energy-dependence will also show up one way or the other, to print the whole outer belt structure. Accelerations for instance will contribute to redistribute the population (e.g. Reeves et al; 2013).

All of this energy dependence is left aside by the authors in their study, maybe because it will be another step, maybe because the measurements are not available from the Meteor satellite (with energy integrated sensors), etc. I understand that it is a limitation of the study that is perfectly admissible (because the authors conclusions are already interesting), however, I would like the authors to have a small paragraph on what they think their limitation is (due to energy-integrated measurement) and they comment the fact that their polar boundary is thus an averaged value of energy-dependent polar boundaries, that may vary quite a lot from the dynamic low energy seed population (~100 keV) up to the high (ultra-relativistic energies, >1 MeV).

About the energy-dependence, there is certainly two populations of the electron outer belts which I am asking the authors to distinguish in their discussions: the low energy electron seed, say around 100 keV, from 50 keV maybe up to 200 keV and higher energies, the core of the outer belt. The seed population is extremely dynamic and will penetrate deep (e.g. Zhao et al. 2013; Turner 2015b, 2016). In other words, it is probably harder to identify both inner and polar boundaries for the seed population. Though I believe that is probably what the authors may be observing. A second issue is that the entering of the seeds population in the outer belt does not occur at its boundary, as, for instance, substorm injections often locates around L=4.5 (Turner 2015b). A third point is that the density of the seed population is much higher than the high energy core, so that they probably dominate once one looks after an integrated outer belt boundary. What the polar boundary ends up to carry in terms of energy population is unknown. Its polar boundary surely does not carry all the information of the outer belt energy structure. The core population is expected to be more stable and, as such, to offer a clearer geometry (but less represented in proportion).

On the contrary, when the polar outer belt boundary is created by the simple effect of the magnetopause via magnetopause incursions and the Dst effect (i.e., magnetopause shadowing), the energy-dependence is totally removed and absent, and, as such, the polar boundary of the outer belts is expected to be this time fully energyindependent. That is expected during disturbed times, that constitutes the second part of the authors' study. Such configuration makes the author's current analysis perfectly valid. It would be good to write it. But again, to observe it, would not it be better to have at disposal some energy resolution (rather than none). That last comment could be a good opening for future work.

In other words, if one tries to locate/study the polar boundary of the outer belts is seems today (with modern technology) more likely to be conclusive if the study is energy-dependent (or at least can be sustained additionally by some energy-dependent measurements), rather than a general global integral for all energies above 100 keV up to 13 MeV, as the authors do. I don't want to minimize the authors results either, as, as I wrote, has definitely its own merit, but I would like to see such opening to be made.

A last link I would like to see to be made is the link between the interactions of the aurora sub-region with the outer belt seed populations through SAPS (Subauroral polarization Streams, e.g. (Kunduri et al., 2017; Lejosne and Mozer, 2017)) potential drop. It has been recently shown that SAPS contribute to the injection (or deeper injections) of the energetic electrons (seed population of the outer belt, up to 200 keV) (Lejosne et al., 2018). This new point has direct implication to the authors' work; 1) it justifies more connecting all aurora associated phenomena and the outer belt dynamics, 2) on the other hand, it explains how fast transport (particularly during disturbed times) will bring more complexity to the outer belt structure, with necessary some implications on its boundaries, 3) it may simply affect the position of their respective boundary, 4) it may open new research angles. Again, it is directly related to the energy dependence of the outer belt, which, as I argue for long enough now here, is a consideration that I would like to see mentioned in this article.

To conclude, I concretely, ask the authors to account briefly for these aforementioned considerations in their discussions with proper citations (full references given below). For instance, Line 24-30 of the conclusions could be a good place for some of these points as the authors already open their discussion to 'acceleration' and 'transport of seed populations' (but I let the authors decide whether it is in the introduction, discussion, conclusion sections).

**References**

Kunduri, B. S. R., Baker, J. B. H., Ruohoniemi, J. M., Thomas, E. G., Shepherd, S. G., & Sterne, K. T. (2017). Statistical characterization of the large-scale structure of the subauroral polarization stream. Journal of Geophysical Research: Space Physics, 122, 6035–6048. https://doi.org/10.1002/2017JA024131

Lejosne, S. & Mozer, F. S. (2017). Subauroral Polarization Streams (SAPS) duration as determined from Van Allen probe successive electric drift measurements. Geophysical Research Letters, 44, 9134–9141. https://doi.org/10.1002/ 2017GL074985

Lejosne, S., Kunduri, B. S. R., Mozer, F. S.,& Turner, D. L. (2018 ). Energeticelectron injections deep into theinner magnetosphere: A result of thesubauroral polarization stream (SAPS)potential drop. Geophysical ResearchLetters, 45, 3811–3819. https://doi.org/10.1029/2018GL077969

Makarevich, R. A., A. C. Kellerman, Y. V. Bogdanova, and A. V. Koustov (2009), Time evolution of the subauroral
electric fields: A case study during a sequence of two substorms, J. Geophys. Res., 114, A04312,

doi:10.1029/2008JA013944.

Reeves, G. D., et al. (2013), Electron acceleration in the heart of the Van Allen radiation belts, Science, 341, 991, doi:10.1126/science.1237743.

Reeves, G. D., et al. (2016), Energy- dependent dynamics of keV to MeV electrons in the inner zone, outer zone, and slot regions, J. Geophys. Res. Space Physics, 121, 397–412, doi:10.1002/2015JA021569.

Ripoll, J.-F., Y. Chen, J. F. Fennell, and R. H. W. Friedel (2014), On long decays of electrons in the vicinity of the slot region observed by HEO3, J. Geophys. Res. Space Physics, 119, doi:10.1002/2014JA020449.

Ripoll, J.-F., O. Santolík, G. D. Reeves, W. S. Kurth, M. H. Denton, V. Loridan, S. A. Thaller, C. A. Kletzing, and D. L. Turner (2017), Effects of whistler mode hiss waves in March 2013, J. Geophys. Res. Space Physics, 122, doi:10.1002/2017JA024139.

Turner, D. L., T. P. O'Brien, J. F. Fennell, S. G. Claudepierre, J. B. Blake, E. K. J. Kilpua, and H. Hietala (2015a), The effects of geomagnetic storms on electrons in Earth's radiation belts, Geophys. Res. Lett., 42, doi:10.1002/2015GL064747.

Turner, D. L., et al. (2015b), Energetic electron injections deep into the inner magnetosphere associated with substorm activity, Geophys. Res. Lett., 42, doi:10.1002/2015GL063225.

Turner, D. L., et al. (2016), Investigating the source of near-relativistic and relativistic electrons in Earth's inner radiation belt, J. Geophys. Res. Space Physics, 121, doi:10.1002/2016JA023600.

Zhao, H., and X. Li (2013), Inward shift of outer radiation belt electrons as a function of Dst index and the influence of the solar wind on electron injections into the slot region, J. Geophys. Res. Space Physics, 118, doi:10.1029/2012JA018179.

**Minor corrections:**

-   P1, Line 10: using the dat
-   P1, Line 28: showed that the polar boundary
-   P2, Line 27: rephrase "with pitch angle lower than 90°" because every p.a. is below 90°…. Do you mean below and close to 90°?
-   P2, Line 28: "due to drift shell splitting (Shabansky effect)". Both are two different things in general. The second can cause the first. Rephrase according to what you want to say. The whole sentence from L25 to L29 is obscure.
-   -P3, line 8: too strong "has not been properly studied yet". You could write something like "still requires careful studies" or "careful examination".
-   P4, Line 4: make sure the definition "of the polar boundary of ORB, also known as the trapping boundary", of the trapping boundary is made the first time you mention the "trapping boundary". (It is not).
-   P7, Line 7: …oval crossing of the ORB. For…
-   P10, line 16: "…ring that surrounds.."
-   P10, line 20: 'It contains closed
-   P10, line 21: 'have drift'
-   P10, line 23: "of the trapping"
-   P10, line 25: feel free to include recent references from the Van Allen Probes observations (e.g. reference above)

---

## Author Response (AR3)

*Referee #1*

**Overall comment**

In this revised manuscript, the authors have addressed all my comments in a satisfactory way. To my mind, the paper is now clear and reaches substantial conclusions of interest for the scientific community. I therefore recommend the paper for publication, provided the few technical corrections given below are made.

**Copyediting and typesetting**

– p. 1, l. 27: high-apogee
– p. 4, l. 1: whether the trapping boundary
– p. 8, l. 11: as separate measures
– p. 10, l. 9: so the increasing of --> so an increase in
– p. 11, l. 2: Investigacin --> Investigación; Tecnologa --> Tecnología

*Dear reviewer 1!*

*Thank you very much for careful reading of our paper and for all corrections!*
*We have made all necessary corrections in the text.*

**ANSWER TO REFEREE 2:**

*Referee #3*

Dear Editor,

This is my review of the paper « Relative locations of the polar boundary of the outer electron radiation belt and the equatorial boundary of the auroral oval » by M. O. Riazanteseva et al. submitted to AnGeo.

This article addresses the problem of finding the position of the polar boundary of the outer electron radiation belt (ORB), relative to the position of the auroral oval (AO). The authors perform a statistical study using the data of the METEOR-M No1 auroral satellite for the period from 11 November 2009 to 27 March 2010. From it they deliver the respective position of the two structures.

I am the third reviewer of this article and I do recommend publications based on the following comments (A) and once the few additional changes I am asking (B) below are made.

A) As a third reviewer, I can see how this article has been improved since its submission. I agree with Reviewer 2 that this article provides a useful scientific step forward (now that some modifications have been made. Reviewer 1 asked to go a bit further in the analysis by trying to determine why d(lat) changes with increasing geomagnetic activity (from totally quiet to moderate activity). This has been done

by the authors through Figure 4 and 5. Figure 4 showing the penetration of the ORB into the AO in terms of delta_Latitude is important. Reviewer 2 asked « new figure(s) that would provide clarity for the reader and confirm that the algorithm is producing results that are consistent with the previous work cited in paragraph 1&2, section 1 ». This has been done; figure 6 showing the distributions of the position of equatorial boundary of the auroral oval (green bins) and the polar ORB boundary (red bins) from the L different AE index is important. The authors have therefore accounted for the changes asked by both reviewers and their final figures are 1- clear to me, 2- bringing a statistical description for different geomagnetic activities, 3- support well the conclusions of the article.

B) There were issues on the energy range for the determination of the ORB and AO. In my review below, I come back on the energy dependence that is a key point to discuss. The main modification I am asking should be fast to do as I only ask 1- to include a small discussion on the energy-dependence of the ORB (with a link to SAPS), 2- to account for recent publications.

My intension is to recommend publication in AnGeo and to see this article published within short delays.

*Dear Reviewer 3!*

*We are grateful to the reviewer for the attention he/she raid to our paper and for a very useful discussion. Unfortunately, the study of the nature of the outer radiation belt and its formation is not the main task of the Meteor satellite mission. Also we have no measurements of electric fields necessary for the analysis of the formation of the ORB. Therefore, it is impossible to add something else, which could be used for the study of the ORB dynamics in addition to the results shown here: the overlapping of the auroral oval and the ORB. We hope that this result is important, and there were very little studies on this subject despite the results of Akasofu [1968].*

*We have tried to analyze all papers relative to this subject and did not find any comparison of the position of the auroral oval with respect to the ORB, made using the data from a single satellite. The observed overlapping of the ORB with the auroral oval raise the question of the auroral oval mapping considering the fact that the drift trajectories of the energetic electrons should be closed around the Earth, and hence cannot be located in the plasma sheet. This result indeed agrees with the latest results (see Antonova et al. [2017, doi:10.1016/j.jastp.2017.10.013] and references in this paper). It became clear that the main part of the auroral oval is mapped to the outer part of the ring current. Such mapping was obtained using a morphological method which does not involve any existing geomagnetic field model, most of which are very overstretched (see [Peredo et al., 1993, JGR, v. 98, No 9., P.*

*15,343-15,354, doi:10.1029/93JA01150; Reeves et al., 1996, Geophys. Mon. 97, p. 167-172, doi:10.1029/GM097p0167 ; Weiss et al. , 1997, JGR.,102, 4911–4918, doi:10.1029/96JA02876; McCollough et al., 2008, Space Weather,6(10), doi:10.1029/2008SW000391; ets.]).* *These new results about the auroral oval mapping made it necessary to re-analyze the relative position of the trapping boundary of energetic electrons with respect to the auroral oval, which was done in our current paper. The results obtained in our study helped us to understand why the previous statistical studies about a relative location of the outer ORB boundary and the equatorial boundary of the auroral oval did not give an unambiguous picture and was interpreted as a coincidence of both boundaries Feldstein and Starkov [1970, PLANET SPACE SCI, 18, 501–508, doi:10.1016/0032-0633(70)90127-3, 1970]).*

Main corrections:

1-: On the energy-dependence
Although authors focus on the polar boundary of the outer belt, I ask here  the question of the impact of energy dependence of the outer belt on its shape, and, therefore, on its boundaries. The inner boundary of the outer belt is extremely energy dependent as shown very recently (e.g. Reeves et al. 2016; Ripoll et al. JGR     2017 and references in them) from the Van Allen Probes, from L~4 to 6. As L-increases above L~6 reaching now the polar boundary of the outer belts, in which the authors are working on, some of this energy-dependence will be kept, in particular for quiet times, for which the plasmasphere extends up to there. From Figure 6 of the authors, it is interesting to see that the probability is higher in the Southern hemisphere. (As a comment, the position of the plasmasphere is determinant on the energy-dependence of the ORB structure, but not only, as wave-particle interactions (WPI) from Chorus waves outside of the plasmasphere will also print an energy-dependence structure of the outer belt). What is interesting about quiet times, on which the authors focus on (but not only), is the great stability of the outer belt. For instance Ripoll et al. 2014 analyzing HEO data shown months of stability of the outer belt for   E>100 keV, reinforcing the relevance of defining outer belt boundaries and positioning them onto known structures (aurora oval, ring plasma or current, plasma sheet, etc.). Still, the stability degrades as L-shell increases above 5-6. (As a comment, there the drift trajectories of the energetic electrons must be closed around the Earth must be more recent observations from the Van Allen Probes, which we let the authors find themselves).   During active times, because WPI     are extremely energy-dependent, it is likely that the energy-dependence      will       also      show up        one       way     or the other, to print the whole outer belt structure.

*Since the discovery of the outer radiation belt by Vernov and his colleagues in 1957 using the data from the second satellite (see the discussion of Baker and Panasyuk [2017, Physics Today, 70(12), 46-51, doi: 10.1063/PT.3.3791]), the dynamics of the ORB was analyzed in many works, including the problem of its stability, and the energy dependence (see figure below from the AE8 model). The AE8 model was developed using the data of high orbiting satellites. New results of the RBSP-Van Allen mission gave new insides about the behavior of electron fluxes (see mentioned papers [Reeves et al., 2016, Ripoll et al., 2017] and multiple other works). It is very difficult to add more for such analysis using the low orbiting Meteor data as ORB electrons are near to isotropic only near ORB polar boundary. Therefore,*

*we consider that better conditions to observe electrons with >100 keV energy at low latitudes in the Southern hemisphere are connected to the north-south asymmetry of the Earth's magnetic field.*

*The estimations of the electron fluxes in the next GGAK-M energy channels (300 and 700 keV) in the outer border of the ORB show that such electron fluxes are not strong enough to be detected by the GGAK-M instrument during quite time (see figure below). Such fluxes can be easily observed only during very disturbed times. Storm-time periods were not analyzed in our study.*

[Figure]

*Fig. Radial profiles of the ORB electron fluxes in accordance with quite time AE8 model at the equatorial plane (thick lines) and at B/Beq=3 (thin lines)*

*Our current work contains the statistics of more than 6200 events of the dropouts to the background level in the electron fluxes with energies more than 100 keV. In the many cases, they were located inside the auroral oval. We did not consider the events of a very strong sharp flux increase at the boundary, because they might be related to the formation of the local particle traps, inside which the energetic electrons can exist during hours [Antonova et al., 2011, J ATMOS SOL-TERR PHY, 73, 1465–1471, doi:10.1016/j.jastp.2010.11.020]. The aforementioned definition is not universal and has its own limitations. In particular, during very quiet geomagnetic conditions the energetic electron fluxes are very low. Therefore, they may reach a defined threshold equatorward from the real ORB boundary. During the analyzed period, the geomagnetic activity was very low. That is why for some events the outer boundary of the ORB was identified to the equator of the equatorial boundary of the auroral oval. We understand that our statistics is sensitive to the energy of threshold, and probably we could have more stable results by elevating its value. Unfortunately, the sensitivity of the high-energy detector onboard the Meteor-M No 1 satellite did not allow us to obtain more accurate results. The new Meteor-*

*M No2 satellite works at much more disturbed geomagnetic conditions. It allows producing measurements for both quiet and disturbed geomagnetic conditions. We plan to continue these studies and obtain how the position of the trapping boundary depends on the energy.*

*The study of ORB stability is also far from the subject of our study in presented research. We agree that it is a very interesting problem connected with constantly observed high level of hiss and chorus waves. Presented by Ripoll et al. [2014] results of the long decay periods of electrons in the slot region are rather interesting in this direction. However, such researches are based on the high apogee satellites and deal with highly anisotropic energetic electron fluxes. In our study we work only with data of low orbiting satellite and electron precipitations not so far from the trapping boundary, where auroral fluxes and energetic electron fluxes are varied in every auroral oval crossing, which is in agreement with Reeves et al. [2016] results.*

Accelerations for instance will contribute to redistribute the population (e.g. Reeves et al; 2013). All of this energy dependence is left aside by the authors in their study may be because it will be another step, may be because the measurements are not available from the Meteor satellite (with energy integrated sensors), etc. I understand that it is a limitation of the study that is perfectly admissible (because the authors conclusions are already interesting), however, I would like the authors to have a small paragraph on what they think their limitation is (due to energy-integrated measurement) and they comment the fact that their polar boundary is thus an averaged value of energy-dependent polar boundaries, that may vary quite a lot from the dynamic low energy seed population (~100keV) up to the high (ultra-relativistic energies, >1 MeV).

*As it was mentioned earlier, in this study we do not analyze the formation of the ORB, which is mainly due to the substorm activity during storms. For instance, we cannot present the comprehensive analysis of energy dependence of the trapping boundary (see above). However comparatively small Larmor radius of energetic electrons - in comparison to the scale of change in the magnetic field - leads to the positioning of all boundaries very close each other in a wide range of energy. The trapping boundary is defined as a boundary which divides the drift trajectories into the trajectories closed inside the magnetosphere and the drift trajectories which intersect the magnetopause. We also know that the maximum in the electron flux of a new radiation belt formed after storm is located at the equatorial boundary of the storm time auroral oval. For example, Reeves et al. [2013] obtained the value L=4 for the 8–9 October 2012 magnetic storm. This storm was later studied by Antonova and Stepanova [2015, EARTH PLANETS SPACE, 67, 148, doi:10.1186/s40623-015-0319-7], who compared the position of the maximum in the electron flux with the position of the maximum in the plasma pressure and the position of the auroral electrojet and found a very good agreement between these positions. During quiet time periods, the auroral oval is mapped to the outer part of the ring current (outer boundary of the oval is located at 10-13 Re according to [Antonova et al., 2017]), therefore we can suggest that the outer boundary of the 1 MeV ORB particles is located inside the auroral oval. However, it is necessary to make additional studies to verify this statement.*

*We added part of current discussion into the new version of paper.*

About the energy-dependence, there is certainly two populations of the electron outer belts which I am asking the authors to distinguish in their discussions: the low energy electron seed, say around 100 keV, from 50 keV maybe up to 200 keV and higher energies, the core of the outer belt. The seed population is extremely dynamic and will penetrate deep (e.g. Zhao et al. 2013; Turner 2015b, 2016). In other words, it is probably harder to identify both inner and polar boundaries for the seed population. Though I believe that is probably what the authors may be observing. A second issue is that the entering of the seeds population in the outer belt does not occur at its boundary, as, for instance, substorm injections often locates around L=4.5 (Turner 2015b). A third point is that the density of the seed population is much higher than the high energy core, so that they probably dominate once one looks after an integrated outer belt boundary.

*Indeed, there is a strong difference between so-called seed population and the electrons with highly relativistic energies [Reeves et al., 2016]. However, electrons with energy >100 keV was considered as a part of ORB (see the figure of AE8 model above). We consider the increase of the seed population as a result of not only deep penetration, but also of the acceleration of electrons in the inner magnetosphere due to auroral processes at low latitudes when the auroral oval moves to low latitudes during magnetic storm, which is in agreement with [Turner et al., 2015b] results. Of course, we agree with the statement that the seed population is extremely dynamic, as it was shown in many works including mentioned Zhao et al. [2013], Turner et al. [2015b, 2016]. It is in a very good agreement with the overlapping between the auroral oval and the outer radiation belt, and with the formation of seed population as a result of substorm activity. It is well established that the substorm expansion phase onset starts from the first auroral arc brightening at the equatorial boundary of the auroral oval (see [Akasofu, 1964] and multiple later works). Dispersionless injection boundary is located at L~6-7 in accordance with [Mauk and Meng, 1983, 88(A12), 10011-10024, doi: 10.1029/JA088iA12p10011] and later works. Shift of the auroral oval to low latitudes during storms leads to the shift of the substorm injection boundary. Therefore, the location of the substorm injections at low latitudes is a very natural result, which is in agreement with the partial overlapping of the auroral oval and ORB. Older popular point of view on the formation of the seed population at the outer boundary of ORB is related to the wrong picture of the auroral oval mapping to the plasma sheet. However, this subject now is not discussed in our paper as it cannot be studied using the quiet time Meteor-M No 1 observations.*

*The aim of our analysis is very simple. We only try to show that the ORB cannot be considered as a region separated from the auroral oval or located at its equatorial boundary. Our statistical study shows that we can see only comparatively small region of the outer part of the ORB where the electron fluxes are nearly isotropic. What is about low latitude injections observed by Turner [2015b], it will be difficult if even possible to study them using low orbiting satellites.*

What the polar boundary ends up to carry in terms of energy population is unknown. Its polar boundary surely does not carry all the information of the outer belt energy structure. The core population is expected to be more stable and, as such, to offer a clearer geometry (but less represented in proportion).

On the contrary, when the polar outer belt boundary is created by the simple effect of the magnetopause via magnetopause incursions and the Dst effect (i.e., magnetopause shadowing), the energy-dependence is totally removed and absent, and, as such, the polar boundary of the outer belts is expected to be this time fully energy-independent. That is expected during disturbed times, that constitutes the second part of the authors' study. Such configuration makes the author's current analysis perfectly valid. It would be good to write it. But again, to observe it, would not it be better to have at disposal some energy resolution (rather than none). That last comment could be a good opening for future work.

In other words, if one tries to locate/study the polar boundary of the outer belts is seems to day (with modern technology) more likely to be conclusive if the study is energy-dependent (or at least can be sustained additionally by some energy-dependent measurements), rather than a general global integral for all energies above 100 keV up to 13 MeV, l as the authors do. I don't want to minimize the authors results either, as, as I wrote, has definitely its own merit, but I would like to see such opening to be made.

*We completely agree with you. The magnetopause shadowing will produce sharp dropouts of all energies. However, this effect is connected to a large increase in the solar wind dynamic pressure. It is necessary to perform a special research to see such dropout in the low orbiting data especially taking into account that our research was done during comparatively quiet conditions. We hope to continue such researches, as was mentioned, using the Meteor-M No 2 satellite.*

*Many problems mentioned by you remain to be solved. We think that the auroral processes usually are not considered as a source of energy for ORB formation, giving the preference to the possible contribution of the chorus waves. Analyzing ORB formation, it is necessary to study the behavior of all energetic particles and this requires the correct calculation of the PSD (phase space density). Naturally, we cannot produce such calculations using low orbiting observations. However, what is important, our analysis indirectly shows that it is necessary to be very careful using the models with predefined current systems - as is commonly used for the RBSP data analysis - as such models, as it was shown, lead, for example, to the distortion in the auroral oval mapping.*

A last link I would like to see to be made is the link between the interactions of the aurora sub-region with the outer belt seed populations through SAPS (Subauroral polarization Streams, e.g. (Kunduri et al., 2017; Lejosne and Mozer,2017)) potential drop. It has been recently shown that SAPS contribute to the injection (or deeper injections) of the energetic electrons (seed population of the outer belt, upto 200 keV) (Lejosne et al., 2018). This new point has direct implication to the authors' work; 1) it justifies more connecting all aurora associated phenomena and the outer belt dynamics, 2) On the other hand, it explains how fast transport (particularly during disturbed times) will bring more complexity to the outer belt structure, with necessary some implications on its boundaries, 3) it may simply affect the position of their respective boundary, 4) it may open new research angles. Again, it is directly related to the energy dependence of the outer belt, which, as I argue for long enough now here, is a consideration that I would like to see mentioned in this article.

*We agree that obtained result on SAPS discovered by Galperin et al. [1973, Kosmicheskie Issledovaniia, 11, 273 - 283] and analyzed in multiple works including latest [Kunduri et al., 2017;*

*Lejosne and Mozer, 2017; Lejosne et al., 2018] is relevant for the ORB and auroral oval partial overlapping. SAPS are observed just to the equator of the equatorial boundary of the auroral oval and are connected to the Region II field-aligned currents of Iijima and Potemra [1976, JGR, 81, 5971-5979, doi: 10.1029/JA081i034p05971]. Estimations presented by [Lejosne et al., 2018] show that the SAPS electric field can be responsible for a deep penetration of "seed" electrons to a slot region. Unfortunately we cannot clarify the role of the subauroral polarization stream on the position of the ORB outer boundary as we have no observations necessary for the analysis of the electric fields. Inside the SAPS regions such fields are very large. However, it is mainly substorm and, naturally, storm phenomena. That is why we think, that our current statistical results are not affected by SAPS, because they were done during very quiet geomagnetic conditions. Formation of SAPS requires comparatively large geomagnetic activity and, naturally, it will be necessary to analyze the effect selected by Lejosne et al. (2018) when it will be possible to deal with magnetic storms.*

*We can stress again that in this study we do not analyze magnetic storms. We only try to show that outer part of the outer radiation belt can be located at latitudes of the auroral oval. Actually, we only carefully support very old, but, unfortunately, forgotten point of view of Akasofu [1968].*

To conclude, I concretely, ask the authors to account briefly for these aforementioned considerations in their discussions with proper citations (full references given below). For instance, Line 24-30 of the conclusions could be a good place for        some of these points as the authors already open their discussion to 'acceleration' and 'transport of seed populations'(but I let the authors decide whether it is in the introduction, discussion, conclusion    sections).

*We add recommended discussion trying not to go far from the subject and aim of our paper. We included the recommended references, considering that although for instance we cannot see direct connections of the discussed by the Reviewer phenomena with the analysis of the relative position of outer ORB boundary and the equatorial boundary of the auroral oval, these connections might appear in the future. We are sure that the study the auroral oval/ORB relations, which can be responsible not only to the appearance of seed population and its dynamics, would lead to many not expected results.*

**Minor corrections:**
- P1, Line 10: using the dat
*Done*

- P1, Line 28: showed that the polar boundary
*Done*

- P2, Line 27: rephrase "with pitch angle lower than 90°" because every p.a. is below 90°…. Do you mean below and close to 90°?
*Yes. We have made the correction.*

- P2, Line 28: "due to drift shell splitting (Shabansky effect)". Both are two different things in general. The second can cause the first. Rephrase according to what you want to say. The whole sentence from L25 to L29 is obscure.
*done*

- -P3, line 8: too strong "has not been properly studied yet". You could write something like "still requires careful studies" or "careful examination".
*done*

- P4, Line 4: make sure the definition "of the polar boundary of ORB, also known as the trapping boundary", of the trapping boundary is made the first time you mention the "trapping boundary". (It is not).
*done*

- P7, Line 7: …oval crossing of the ORB. For…
*done*

- P10, line 16: "…ring that surrounds.."
*done*

- P10, line 20: 'It contains closed
*done*

- P10, line 21: 'have drift'
*done*

- P10, line 23: "of the trapping"
*done*

- P10, line 25: feel free to include recent references from the Van Allen Probes observations (e.g. reference above)
*done*

**All relevant changes made in the manuscript are shown by blue color in the marked-up manuscript version below.**

[revised manuscript text omitted]